# Giant chiral magnetoelectric oscillations in a van der Waals multiferroic

Frank Y. Gao[1,9], Xinyue Peng[1,9], Xinle Cheng[2], Emil Viñas Boström[2,3], Dong Seob Kim[1], Ravish K. Jain[4], Deepak Vishnu[4,5], Kalaivanan Raju[4], Raman Sankar[4], Shang-Fan Lee[4], Michael A. Sentef[2,6], Takashi Kurumaji[7], Xiaoqin Li[1], Peizhe Tang[2], Angel Rubio[2,3,8 ✉] & Edoardo Baldini[1 ✉]

Helical spin structures are expressions of magnetically induced chirality, entangling the dipolar and magnetic orders in materials[1–4]. The recent discovery of helical van der Waals multiferroics down to the ultrathin limit raises prospects of large chiral magnetoelectric correlations in two dimensions[5,6]. However, the exact nature and magnitude of these couplings have remained unknown so far. Here we perform a precision measurement of the dynamical magnetoelectric coupling for an enantiopure domain in an exfoliated van der Waals multiferroic. We evaluate this interaction in resonance with a collective electromagnon mode, capturing the impact of its oscillations on the dipolar and magnetic orders of the material with a suite of ultrafast optical probes. Our data show a giant natural optical activity at terahertz frequencies, characterized by quadrature modulations between the electric polarization and magnetization components. First-principles calculations further show that these chiral couplings originate from the synergy between the non-collinear spin texture and relativistic spin–orbit interactions, resulting in substantial enhancements over lattice-mediated effects. Our findings highlight the potential for intertwined orders to enable unique functionalities in the two-dimensional limit and pave the way for the development of van der Waals magnetoelectric devices operating at terahertz speeds.

Chirality—the property of an object to be non-superimposable on its mirror image—is a fundamental concept in the natural world. It manifests itself across a vast spectrum of systems, from the spirals of galaxies in the cosmos to the helical structure of DNA in living organisms. In crystalline solids, chirality typically emerges from the asymmetry of spatial coordinates among constituent atoms in systems lacking axes of improper rotation. This geometric chirality underlies myriad phenomena, including optical circular dichroism[7], unusual photogalvanic effects[8] and nonreciprocal quantum transport[9]. Beyond this lattice-mediated chirality, other forms of chirality exist in solids with otherwise achiral lattices, generated by instabilities such as charge-density waves[10], gyrotropic orders[11] and spin-spiral magnetic textures[12–14] (Fig. 1a). The latter, in particular, are gaining importance because of their ability to drive macroscopic ferroelectric polarizations that are intertwined with the underlying magnetic order and hold relevance to a wide range of technological applications. Materials realizing this physical regime of enhanced magnetoelectric couplings are known as type-II multiferroics[1–4].

Recently, a concerted effort has been made to extend type-II multiferroics into the atomically thin regime by exploiting van der Waals materials that crystallize in centrosymmetric structures[5,6]. These quasi-two-dimensional multiferroics have the potential to achieve enhanced magnetoelectric properties with a level of tunability that is beyond the reach of traditional three-dimensional helimagnets. At finite frequency, these properties are expected to be controlled by different manifestations of the dynamical magnetoelectric coupling, expressed by the first-order magnetoelectric response[12] $\Delta P_i(\omega) = \alpha_{ij}(\omega)\Delta H_j(\omega)$. In this relation, $\alpha_{ij}(\omega)$ is the complex dynamical magnetoelectric susceptibility tensor, which represents the cross-coupling between the modulation of the electric polarization $\Delta P_i(\omega)$ and magnetizing field $\Delta H_j(\omega)$ of the material. Among the various phenomena governed by this tensor, the natural optical activity stands out as the effect most intimately connected to the spontaneous breaking of chiral symmetry[13,14]. This property describes the rotation of light polarization that is symmetric under time reversal and antisymmetric under spatial inversion, and its strength can be enhanced in resonance with the collective modes of the magnetochiral order[14]. Showing the presence and magnitude of this unknown effect in the realm of exfoliated van der Waals multiferroics is pivotal for realizing high-speed chiral applications based on artificially stacked layered structures. However, the lack of suitable probing methods has so far hindered the identification of this phenomenon at the microscopic length scales of individual chiral domains in a van der Waals flake.

[1]Department of Physics and Center for Complex Quantum Systems, The University of Texas at Austin, Austin, TX, USA. [2]Max Planck Institute for the Structure and Dynamics of Matter, Hamburg, Germany. [3]Nano-Bio Spectroscopy Group, Departamento de Física de Materiales, Universidad del País Vasco, San Sebastián, Spain. [4]Institute of Physics, Academia Sinica, Taipei, Taiwan. [5]Department of Chemistry, National Tsing Hua University, Hsinchu, Taiwan. [6]Institute for Theoretical Physics and Bremen Center for Computational Materials Science, University of Bremen, Bremen, Germany. [7]Division of Physics, Mathematics and Astronomy, California Institute of Technology, Pasadena, CA, USA. [8]Center for Computational Quantum Physics, The Flatiron Institute, New York, NY, USA. [9]These authors contributed equally: Frank Y. Gao, Xinyue Peng. ✉e-mail: angel.rubio@mpsd.mpg.de; edoardo.baldini@austin.utexas.edu

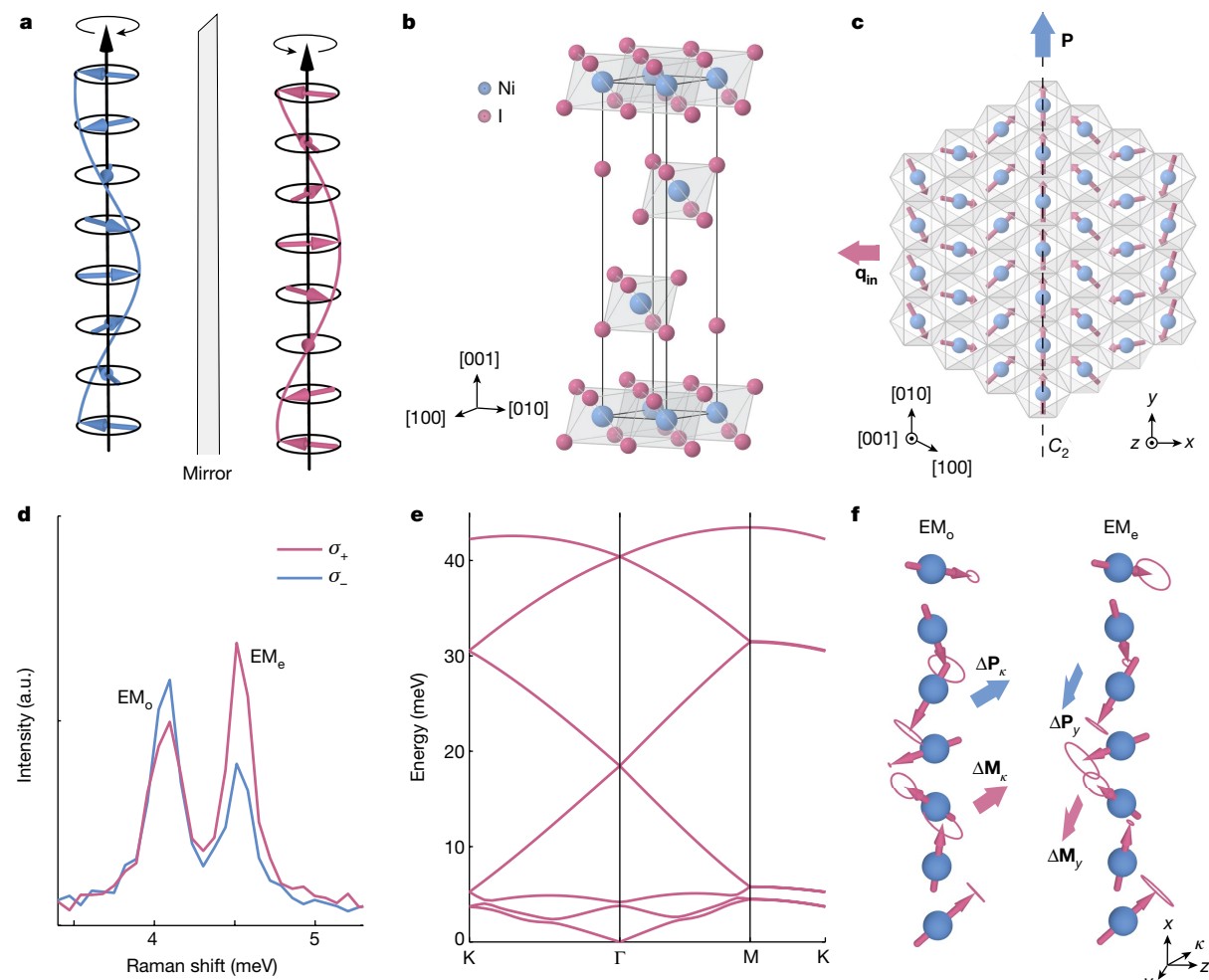

**Fig. 1 | Crystal structure and electromagnon modes of NiI₂. a**, Spin-spiral ordering exemplified by two spin helices with opposite left and right handedness. **b**, The rhombohedral crystal structure of NiI₂ at high temperature. The nickel (blue) and iodine (red) atoms are arranged in triangular lattice layers. **c**, The spin-spiral order depicted on the (001) plane below $T_{HM} \approx 60$ K with propagation vector **q** = (0.138$a^*$, 0, 1.457$c^*$) and in-plane component $\mathbf{q}_{in}$, and electric polarization **P** along $y$, parallel to the $C_2$ axis. Also shown is the $x$, $y$, $z$ coordinate system, where $y \parallel [010]$ and $z \parallel [001]$. **d**, Low-frequency spontaneous Raman scattering measurement of exfoliated NiI₂ with $\sigma_+$ (red) and $\sigma_-$ (blue) circularly polarized light obtained at 2.4 K. The two peaks at 4.09 meV and 4.51 meV are the EM₀ and EMₑ electromagnons, respectively. **e**, Energy–momentum dispersion of the electromagnon modes calculated by DFT on a 7 × 1 unit cell of monolayer NiI₂. **f**, Eigenvector spin precessions of EM₀ and EMₑ, shown alongside the calculated $\Delta\mathbf{P}$ and $\Delta\mathbf{M}$ vectors for each mode. a.u., arbitrary units.

To address this challenge, here we perform a precision measurement of the dynamical magnetoelectric coupling for a native enantiopure domain of the van der Waals multiferroic nickel(II) iodide (NiI₂)[15]. We use a tailored laser pulse to coherently drive an electrically active magnon mode and directly track the resulting modulation of the dipolar and spin-spiral orders in real time using optical second-harmonic generation (SHG) and Kerr rotation microscopy. Our protocol shows the existence of natural optical activity at terahertz frequencies, signalled by the emergence of a π/2 phase shift between the electric polarization and magnetization oscillations. Extraction of the corresponding magnetoelectric tensor element shows that the strength of natural optical activity greatly surpasses that of any other known helimagnet. Our comprehensive analysis, which incorporates tight-binding model, density-functional theory (DFT) and group theory calculations, fully rationalizes these findings, underscoring the role of the relativistic spin–orbit interaction in the origin of such giant natural optical activity.

The layered NiI₂ crystallizes in a CdCl₂-type rhombohedral lattice with space group $R\bar{3}m$ (ref. 15). In this solid, the Ni²⁺ (S = 1) ions are surrounded by an octahedral I⁻ environment and form a triangular lattice, resulting in substantial magnetic frustration (Fig. 1b). On cooling under zero magnetic field, the system undergoes two successive phase transitions, first from a paramagnetic to a collinear antiferromagnetic state at $T_{AFM} \approx 75$ K, and subsequently to a helimagnetic phase at $T_{HM} \approx 60$ K. The latter is characterized by a proper-screw spin order with propagation vector **q** = (0.138$a^*$, 0, 1.457$c^*$) in reciprocal lattice units[16]. This spiral order breaks inversion, mirror and the $c$-axis three-fold rotational symmetry, leading to a monoclinic structure that retains an in-plane $C_2$ axis. The inverse Dzyaloshinskii–Moriya (IDM) mechanism and spin-dependent metal–ligand hybridization further induce an in-plane electric polarization perpendicular to **q** (refs. 6,15,17,18) (Fig. 1c). The net result is the emergence of a type-II multiferroic state that persists down to the ultrathin limit[5,6] and hosts a pair of low-energy electromagnons exhibiting both Raman and infrared activity at the Brillouin zone centre[6,19,20]. In a previous study[6], it was discovered that these modes exhibit large Raman circular dichroism and a complex set of polarization selection rules, which is a direct manifestation of the underlying chirality of the spin order. In our Raman measurements performed under circularly polarized light, the electromagnons appear at 4.09 meV and 4.51 meV around 2.4 K (Fig. 1d).

## Theoretical modelling

We first gained theoretical insights into the strength of the magneto-electric coupling in resonance with these electromagnon modes. To this aim, we conducted DFT calculations and obtained the magnetic exchange parameters of the spin Hamiltonian describing $NiI_2$ (Methods). Our model predicted the expected proper-screw spin order, stabilized by the competition between a ferromagnetic nearest-neighbour exchange and an antiferromagnetic third nearest-neighbour exchange[17]. Within this spin-spiral phase, an in-plane electric polarization emerged perpendicular to the proper-screw vector, exhibiting a magnitude consistent with experimentally reported values[15]. We then computed the magnon dispersion by solving the Landau–Lifshitz equation in a $7 \times 1$ supercell, which best captures the non-commensurate spiral periodicity[17]. The results, shown in Fig. 1e, indicated the formation of several zone-folded magnon modes in the helimagnetic phase, among which are a pair of zone-centre $C_2$-odd ($EM_o$) and $C_2$-even ($EM_e$) excitations at 3.92 meV and 4.30 meV, respectively. The spin precessions associated with these two modes are shown in Fig. 1f. To understand the magnetoelectric response of these collective excitations, we examined their electric and magnetic dipole moments within the frozen magnon approximation[21], supplementing our first-principles theory with model Hamiltonian calculations[22]. This approach allowed us to determine that the modes involve modulations of the polarization ($\Delta\mathbf{P}$) and magnetization ($\Delta\mathbf{M}$) that are either parallel ($EM_e$ along $\hat{y}$) or perpendicular ($EM_o$ along $\hat{k}$) to the $C_2$-axis as shown in Fig. 1f. By considering these modulations along with the relation $\Delta P_i(\omega) = -\alpha_{ij}(\omega)\Delta M_j$, that is, the limit of the classical magnetoelectric response in the absence of external fields, we extracted the magnetoelectric coupling strength in resonance with the two electromagnons. This analysis showed colossal values of $\mathrm{Im}[\alpha_{ii}] \approx 10^4$ ps m$^{-1}$ for both modes (Extended Data Table 1) and clarified that the coupling strengths in $NiI_2$ originate from the interplay among an electronically driven non-collinear magnetic order, a substantial spin–orbit interaction on the ligand atoms and a strong $d$–$p$ hybridization[22]. The imaginary component of the diagonal tensor elements—a signature of natural optical activity[13]—is a consequence of the quasi-time-reversal symmetry of the material, that is, a combination of time-reversal and translation symmetry (Supplementary Note 2).

## Ultrafast probes of electromagnon dynamics

To experimentally validate these predictions, we measured $NiI_2$ flakes with a typical lateral size of 30 μm × 30 μm and a thickness of about 100 nm. Following mechanical exfoliation and transfer into our cryostat, the samples contained a racemic mixture of chiral helimagnetic domains below $T_{HM}$. Initially, we identified an enantiopure domain with defined chirality using SHG microscopy. Specifically, we probed the SHG polarimetry pattern using photon energy that was off-resonant with $d$–$d$ transitions (Extended Data Fig. 1). This probing configuration is sensitive only to the underlying ferroelectric order (Supplementary Note 3), which, in $NiI_2$, is spatially intertwined with the magneto-chiral order. Tracking the SHG polarimetry response as a function of temperature showed a sharp increase in the signal intensity upon formation of the multiferroic phase, accompanied by a transition from a six-fold- to two-fold-symmetric polar pattern (Fig. 2a). Notably, the latter matches the electric-dipole SHG of a material with point group 2 and $C_2$-axis oriented along the azimuthal angle $\phi = 90°$, consistent with the emergence of a ferroelectric polarization from a single domain. After isolating the enantiopure domain, we perturbed the multiferroic state with a laser pulse, triggering coherent oscillations of the electromagnon modes. To minimize transient heating, we selected our pump photon energy to lie below the charge-transfer gap of the material, in a spectral region between different $d$–$d$ transitions[23–27]. As most pump photons remained unabsorbed, the pump beam acted only as a weak perturbation to the ground state. Leveraging the dual infrared and Raman activity of the electromagnons, we then used a weak, off-resonant probe pulse and monitored the coherent oscillations of the mode in the SHG and Kerr rotation channels[28,29]. As detailed in Supplementary Note 5, these probes are, respectively, sensitive to the polarization and magnetization modulations of the material.

Figure 2b shows the coherent oscillatory response in the time-resolved SHG (tr-SHG) channel as a function of temperature in the cross-polarized configuration. For these measurements, we chose the incident probe polarization to be perpendicular to the $C_2$-axis ($\phi = 0°$). Under these conditions, the response is well described by the dynamics of a single collective mode, which both weakens and softens with increasing temperature. A fit of the temporal traces with a single damped sinusoidal function shows that this excitation coincides with $EM_o$ (Extended Data Fig. 2). This is expected because fluctuations of this mode lead to modulations of the electric polarization perpendicular to the $C_2$-axis. To visualize the full coherent dynamics of the electric polarization on the $NiI_2$ planes, we measured the anisotropic SHG response while varying the incident probe polarization in the (001) plane[30–32]. Figure 2c–f shows the data for both parallel- and cross-polarized configurations. By examining the time-dependent polar patterns (Fig. 2c,e), we observed that the parallel-polarized signal showed a two-fold pattern modulated by a symmetry-conserving mode, whereas the cross-polarized signal exhibited symmetry-breaking modulations that result in alternating four-lobe and two-lobe patterns. The former is consistent with the oscillation of a single mode, whereas the latter suggests a complex interference between both modes. This interpretation is further supported by the tr-SHG polar maps in Fig. 2d,f. Here the oscillations in the parallel configuration show a stable phase relation across both polarization and temporal axes, indicative of a single-mode response. By contrast, the crossed-polarized configuration signal contains oscillations that gradually dephase from one another, a behaviour suggestive of a beating between both electromagnons.

We retrieved the time-dependent evolution of the second-order susceptibility tensor elements by modelling the anisotropic tr-SHG signal (Supplementary Note 4). The results are shown in Fig. 3a. The extracted susceptibilities can be partitioned into two groups (shown in blue and red), oscillating at the $EM_e$ and $EM_o$ energies (Fig. 3b). Note that the observed energies are slightly softened compared with those in the static Raman spectra because of the residual pump absorption (Supplementary Fig. 14). Taking the proportionality between the susceptibility modulations and the polarization, we could track the evolution of the latter in the $xy$ plane (Fig. 3c). The spiral pattern that emerges with shifting ellipticity originates from the beating between the two coherent modes dephasing from one another.

After probing the dynamical evolution of ferroelectricity in the SHG channel, we monitored the corresponding magnetization dynamics using time-resolved reflective Kerr rotation (tr-RKerr)[33]. Figure 3d shows the oscillatory signals measured in a polar tr-RKerr geometry for different pump fluences. We obtained an excellent fit for these traces with a model based on a single damped harmonic oscillator. As shown in Fig. 3b, the extracted electromagnon energy matches the one of $EM_o$. By contrast, contributions from $EM_e$ seem to be weak or absent in the tr-RKerr signal for the given in-plane pump and probe polarization. The simultaneous presence of the coherent $EM_o$ response in the tr-SHG and tr-RKerr channels under identical experimental conditions enabled us to establish a precise comparison between the parameters of the oscillations (see Supplementary Note 5). This comparison is crucial for explaining the nature of the phenomenon governing the underlying magnetoelectric response. Figure 3e shows the initial phase of the oscillations as a function of pump fluence. We observed a striking $\pi/2$ phase shift in the $EM_o$ mode oscillations between the two channels across all fluence values. This phase difference aligns with our DFT and spin model calculations, which predict that the electric polarization and magnetization evolve in quadrature with each other (Supplementary

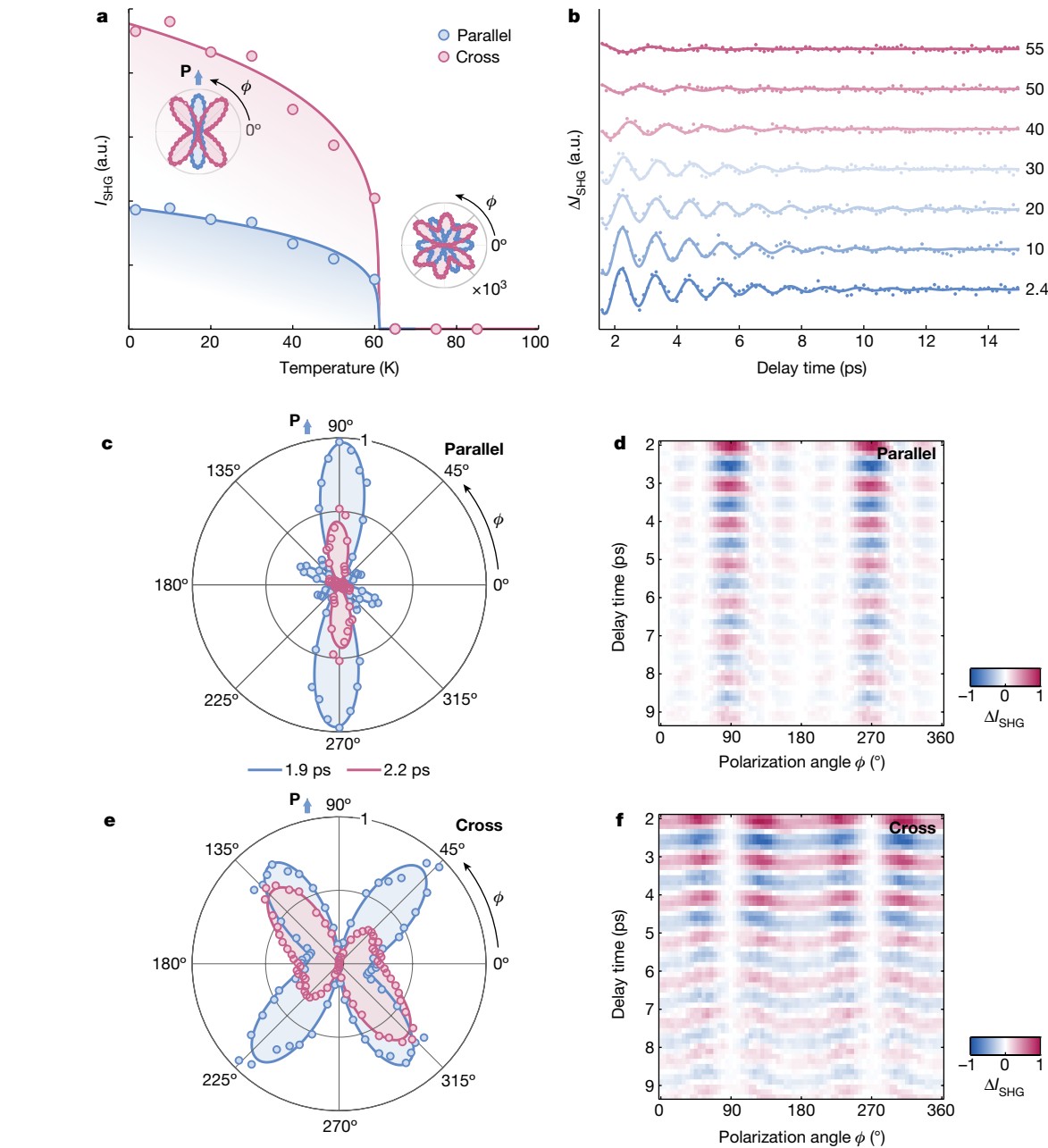

**Fig. 2 | Static and transient SHG polarimetry. a**, Temperature-dependent static SHG signal on an enantiopure domain of exfoliated $NiI_2$ collected in both parallel- (blue) and cross- (red) polarized detection configurations. The insets show the anisotropic SHG signal both above and below the transition temperature $T_{HM} \approx 60$ K. **b**, Temperature-dependent oscillatory tr-SHG measurements on $NiI_2$ taken at an incident pump fluence of 1.10 mJ cm$^{-2}$ with the probe perpendicular to the $C_2$-axis ($\phi = 0°$) in the cross-polarized configuration. The oscillator energy matches the $C_2$-odd electromagnon, $EM_o$

(Extended Data Fig. 2). **c**,**e**, The anisotropic tr-SHG signal at a pump–probe delay time of 1.9 ps (blue) and 2.2 ps (red) for both parallel- (**c**) and cross-polarized (**e**) detection configurations. **d**,**f**, The anisotropic tr-SHG signal is shown as a function of pump–probe delay time and the probe polarization angle $\phi$ for both parallel- (**d**) and cross-polarized (**f**) detection configurations. The data were collected with an incident pump fluence of 1.65 mJ cm$^{-2}$ at 2.4 K. a.u., arbitrary units.

Note 2). The same models also explain the absence of the $EM_e$ coherent signal in the polar tr-RKerr response, attributing it to the $C_2$-even electromagnon lacking an out-of-plane magnetization component. Thus, our findings underscore the distinct sensitivities of tr-SHG and tr-RKerr probes to variations in dipolar and magnetic order, respectively.

## Dynamical magnetoelectric coupling

Finally, we retrieved the magnetoelectric coupling constant for $EM_o$ by transforming the corresponding tr-SHG and tr-RKerr signals to

modulations of the material polarization $\Delta P_\kappa$ and magnetization $\Delta M_\kappa$ projected onto the corresponding $\kappa$ axis. We performed this conversion for oscillatory signals at a pump fluence of 1.65 mJ cm$^{-2}$ (Supplementary Note 6). The resulting dynamics are shown in Fig. 4a (solid lines) alongside the theoretical prediction (shaded curve). Importantly, the $\pi/2$ out-of-phase response characterizes both curves, confirming that the observed magnetoelectric effect is related to chiral symmetry breaking and natural optical activity. We also obtain a giant magnetoelectric coupling constant along the $EM_o$ modulation axis $\kappa$ of $\mathrm{Im}\{\alpha_{\kappa\kappa}\} = 11 \times 10^3$ ps m$^{-1}$ at around 4 meV (that is, 1 THz), in excellent

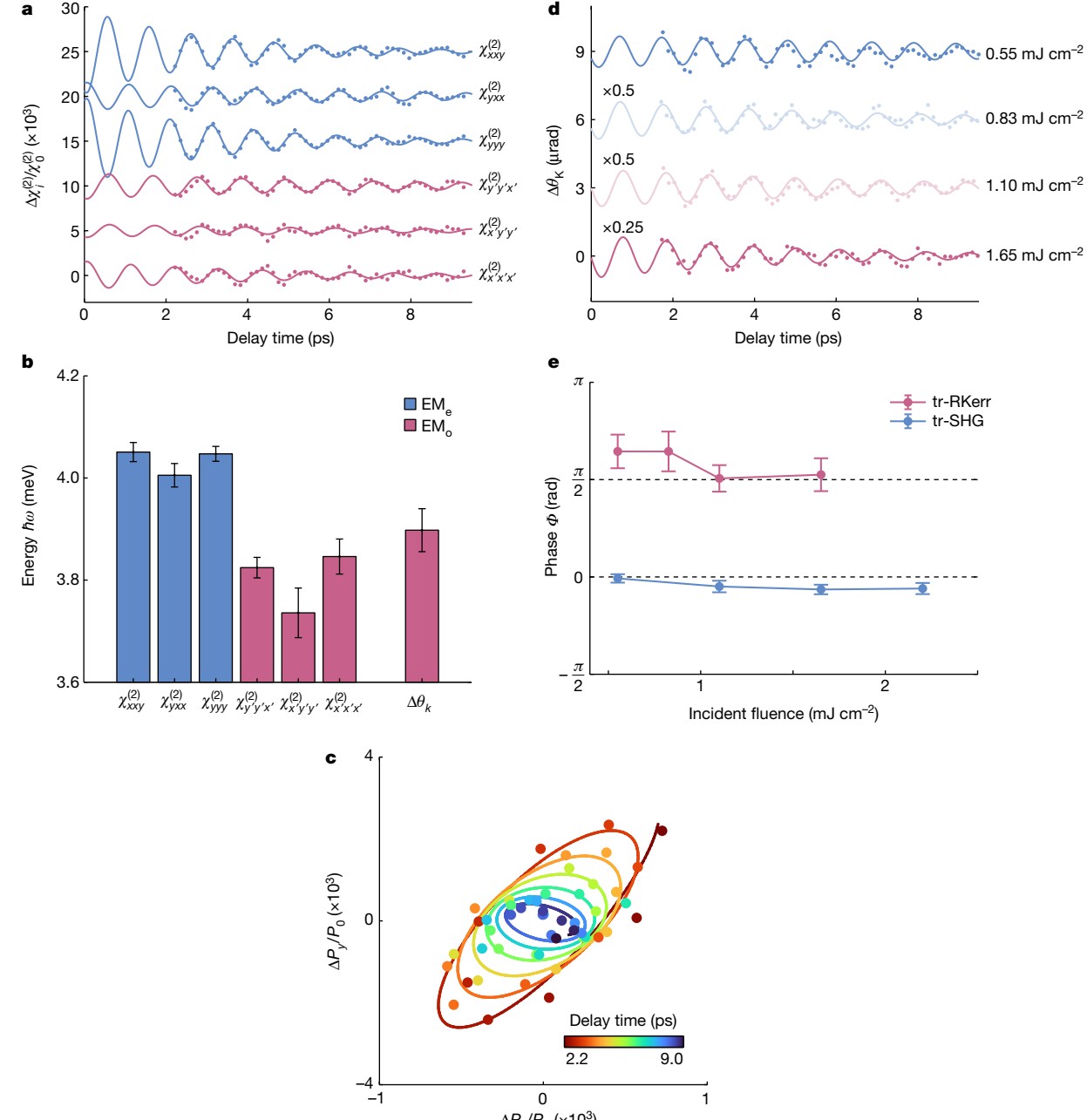

**Fig. 3 | Electric polarization and magnetization dynamics. a**, The second-order nonlinear optical susceptibilities $\Delta\chi^{(2)}_{ijk}(t)$ (dots), extracted from Fig. 2d,f, are shown as a function of time along with fits (lines) to a damped oscillator model. The curves have been vertically offset for clarity. **b**, Oscillator energies obtained from the fits indicate separate electromagnon modulations of the $C_2$-even-like (blue) and $C_2$-odd-like (red) tensor elements. The frequency obtained from the tr-RKerr signal $\Delta\theta_k$ (see **d**) at the same pump fluence is also shown. **c**, Temporal evolution of the electric polarization components $\Delta P_x$ and $\Delta P_y$ induced by the oscillations of both electromagnon modes compared with the static polarization $P_0$. Data points (dots) are shown alongside fits of the temporal dynamics (lines). **d**, tr-RKerr signals for several incident pump fluences at 2.4 K. Experimental data (dots) are shown alongside fits to a damped oscillator model (lines). The curves have been vertically offset for clarity. **e**, The initial oscillatory phases, $\Phi$, obtained from the fits of the tr-SHG and tr-RKerr signals are shown as a function of incident pump fluence (Supplementary Note 5). All error bars show the 95% confidence intervals obtained from the fits.

agreement with our theoretical value of $12 \times 10^3$ ps m$^{-1}$. This, in turn, yields a large natural optical activity ($\eta \approx 1,000°$ mm$^{-1}$) surpassing that of other helimagnets, including CuO (ref. 14) and CuFe$_{0.965}$Ga$_{0.035}$O$_2$ (ref. 13). This value also exceeds the strength of other magnetoelectric coupling phenomena mediated by the same tensor (for example, gyrotropic birefringence and directional dichroism) observed in all measured single-phase multiferroics[12,34–36] (Fig. 4b and Supplementary Table 7).

## Discussion

Our theoretical model explains the reasons for the pronounced natural optical activity exhibited by NiI$_2$ at terahertz frequencies (Supplementary Note 2). In contrast to other multiferroics in which the electromagnons stem from phonon-mediated mechanisms, in NiI$_2$, these excitations originate from the IDM interaction. This purely electronic character conspires with the spin–orbit coupling on the ligands and the

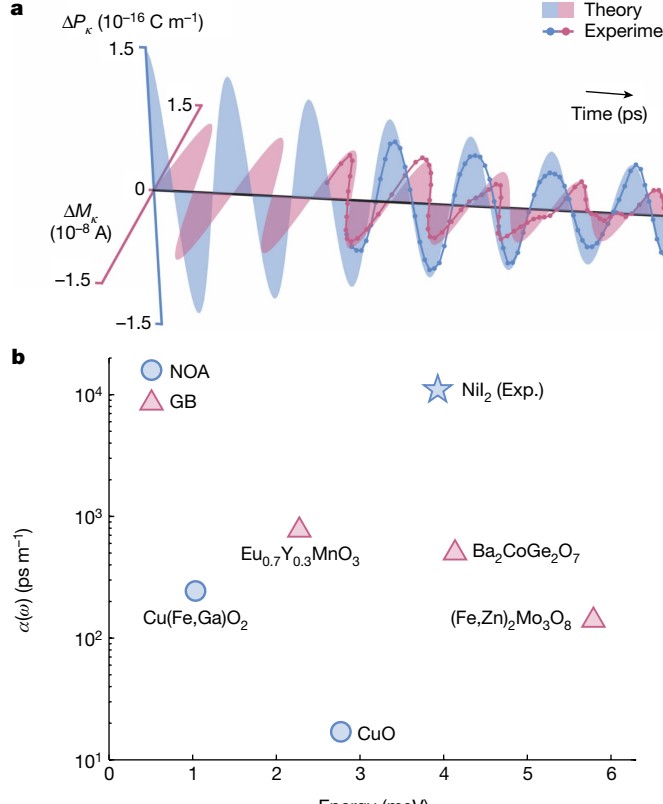

**Fig. 4 | Electromagnon dynamical magnetoelectric coupling.**
**a**, Time-dependent modulations of the two-dimensional electric polarization and magnetization extracted for $EM_o$ from the tr-SHG (blue dots, lines) and tr-RKerr measurements (red dots, lines). Theoretical electric polarization and magnetization dynamics of the $EM_o$ mode obtained from DFT calculations and presented assuming a phenomenological damping rate (filled areas). To clarify the oscillatory dynamics, the experimental data were interpolated by a factor of 2 and then smoothed with a moving average filter. **b**, The dynamical magnetoelectric coupling constant $\alpha(\omega)$ in the terahertz spectral range retrieved in a variety of materials (Supplementary Table 7). Data points are categorized by their associated optical rotational phenomena. NOA, natural optical activity (blue circles); GB, gyrotropic birefringence (red triangles).

$d-p$ hybridization between the nickel and the iodine atoms, resulting in a giant electric polarization and consequent magnetoelectric coupling strength. In particular, the $d-p$ hybridization in $NiI_2$ is unusually strong because of the substantial electronegativity of nickel and the large atomic radius of iodine[22]. Collectively, our findings underscore the extraordinary magnetoelectric performance that is attainable through direct electronic interactions, providing a roadmap for the realization of even larger chiral magnetoelectric couplings in future spiral magnets. We believe that the layered nature of van der Waals solids could lead to further enhancements of chirality by spin–orbit coupling engineering at interfaces[37] or tailored suppression of electronic screening[38].

## Conclusion

In conclusion, we have observed giant terahertz magnetoelectric oscillations from an individual chiral domain in a van der Waals multiferroic. Our study has established the key factors necessary to achieve unparalleled dynamical magnetoelectric interactions, which could play a crucial part in next-generation devices for chiral spintronics and sensing. Moreover, the extraction of the phase of the electromagnon oscillations realized in our approach will enable chirality-sensitive domain imaging in various multiferroics. Extensions of our theoretical model to account

for nonlinear magnonic couplings[39,40] and light–matter hybridization would show the effects of chiral magnetoelectric correlations in the presence of intense driving fields or polaritonic environments. Assessing these challenges could pave the way for the coherent switching of chiral domains by intense terahertz pulses[41–43] and the emergence of exotic magnetic states mediated by chiral cavity fields[44–46].

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

## Methods

### NiI$_2$ single-crystal growth

NiI$_2$ single crystals were grown in a similar method as previously reported[16]. Pure nickel (99.99%) (Ni) and iodine (I$_2$) reagents were purchased from Sigma Aldrich. The growth of NiI$_2$ single crystals was performed in quartz ampoules by mixing Ni and I$_2$ powders under a vacuum of $10^{-3}$ Pa. The ampoules were placed in a horizontal furnace at 650 °C for 2 days to obtain partially mixed crystals. Afterwards, the mixed crystals were used to grow the pure NiI$_2$ single crystals with the help of a chemical vapour transport method setup. For the growth of NiI$_2$ single crystals, the end of the ampoule containing the preheated material was held at 750 °C, and the growth end was maintained at a temperature near 650 °C with a temperature gradient near 2.5 °C cm$^{-1}$ for 2 weeks. Shiny NiI$_2$ single crystals of sizes up to 6 × 4 × 1.5 mm$^3$ were obtained.

### Sample preparation and optical measurements

NiI$_2$ flakes with bulk-like thickness were mechanically exfoliated with scotch tape and polydimethylsiloxane (PDMS) onto Si substrates covered with a 285-nm-thick SiO$_2$ layer. The preparation was done in an N$_2$-filled glovebox to prevent sample degradation. To minimize exposure to ambient air, the sample transfer from the glovebox to the cryostat was carefully managed, ensuring an exposure time of less than 5 min. The measured flakes have a typical lateral size of 30 μm × 30 μm and a thickness of about 100 nm.

All optical measurements, including spontaneous Raman scattering, SHG and RKerr, were performed within a helium-cooled closed-cycle cryostat (Quantum Design, OptiCool) with a temperature range from 1.6 K to 350 K. Nanopositioners (Attocube, ANPx101/LT-linear x-nanopositioner) with sub-nanometre precision were used to control the sample position. To focus the light onto the sample and collect the optical signal, a 50× objective (Mitutoyo MY50X-825, numerical aperture 0.42) was used. In the time-resolved experiments, the pump and probe beams were combined with a beamsplitter. They propagated collinearly into the objective and overlapped on the sample. To ensure that the pump spot size was larger than the probe spot size, the probe photon energy (1.20 eV) fell within the achromatic range of our objective (1.16–2.67 eV), whereas the pump photon energy (1.13 eV) did not. Thus, the probe was tightly focused into a diffraction-limited spot of 1 μm onto the sample, whereas the pump spot size remained poorly focused and relatively large (approximately 2.5 μm). This approach enabled us to create a uniform photoexcited region to be probed.

### Spontaneous Raman scattering

Circularly polarized spontaneous Raman scattering measurements were performed using a 2.33 eV (532 nm) continuous wave laser (Cobolt Samba) with an incident power of 150 μW. The inelastically scattered beam was collected using a mid-focal-length imaging spectrometer (Horiba, iHR550) equipped with a diffraction grating of 1,800 grooves per mm. The Raman signal was detected with a liquid-nitrogen-cooled charged-coupled device camera (Pylon 100BR eXcelon). An achromatic quarter-wave plate was used to generate left and right circular polarized light for Raman circular dichroism measurements (Fig. 1d). Each scan was integrated over 6 min, and no analyser was used.

### Static and time-resolved optical second-harmonic generation

For static SHG measurements, an amplified Yb:KGW laser (Light Conversion, Carbide 40 W) was used to generate probe pulses centred around 1.20 eV at a 100 kHz repetition rate. A motorized half-wave plate was inserted to rotate the incoming probe polarization and collect SHG polarimetry patterns. The back-reflected SHG beam was directed to an analyser, which selected the SHG signal parallel- or cross-polarized with the incident probe light. After filtering out the fundamental light, the SHG signal was detected by a photomultiplier tube (Hamamatsu, H9305-01) coupled to a lock-in amplifier (Zürich Instruments, UHFLI).

For tr-SHG experiments, a portion of the laser was coupled into an optical parametric amplifier (Light Conversion, Orpheus HF) to generate pump pulses centred around 1.13 eV. The pump repetition rate was set at 50 kHz using an electro-optic modulator (ConOptics, M350). The pulse duration of the pump and probe pulses was 100 fs and 270 fs, respectively. Full tr-SHG polar maps were collected by varying the incident probe polarization and the pump–probe delay, with the latter done using a delay-line stage.

### Time-resolved reflective Kerr rotation

tr-RKerr measurements were performed using the same pump and probe pulses as in the tr-SHG experiments. The Kerr rotation induced by the pump was detected by a polarization-resolved scheme. In this setup, the reflected probe light was split into two beams polarized at ±45° with respect to the incident polarization using a Wollaston prism and a half-wave plate. These orthogonally polarized beams were directed into a balanced amplified photodetector (Thorlabs PDB230A). The differential pump-induced photocurrent was then read out by a lock-in amplifier synced to the 50 kHz pump modulation frequency.

### Effective spin Hamiltonian

We studied the magnetic properties of NiI$_2$ under the assumption that interlayer interactions can be neglected. The resulting Hamiltonian of a single magnetic layer is $H = H^{iso} + H^{ani}$, where the dominant isotropic part $H^{iso}$ is

$$H^{iso} = \frac{1}{2} \sum_{ij} J_{ij} \mathbf{S}_i \cdot \mathbf{S}_j + \frac{B}{2} \sum_{\langle ij \rangle} (\mathbf{S}_i \cdot \mathbf{S}_j)^2, \tag{1}$$

where $J_{ij}$ are isotropic exchange interactions extending up to the third nearest neighbours, denoted respectively by $J_1$, $J_2$ and $J_3$. The dominant term $J_1 = -5.03$ meV is a ferromagnetic nearest-neighbour exchange, complimented by a small antiferromagnetic next nearest-neighbour interaction $J_2 = 0.32$ meV and a large antiferromagnetic third nearest-neighbour exchange $J_3 = 3.95$ meV. The competition between these interactions results in a proper-screw spin-spiral ground state[47]. Finally, $B = -0.89$ meV is a ferromagnetic nearest-neighbour biquadratic exchange, which acts as a refinement to $J_1$.

Apart from the dominant isotropic Hamiltonian, we find an anisotropic correction of the form

$$H^{ani} = \frac{1}{2} \sum_{ij} \mathbf{S}_i \cdot (J_{ij}^r \mathbf{S}_j) + \frac{1}{2} \sum_i \mathbf{S}_i \cdot (A_s \mathbf{S}_i), \tag{2}$$

where $J_{ij}^r$ is a traceless and symmetric matrix describing nearest-neighbour ($J_1^r$) and third nearest-neighbour ($J_3^r$) anisotropic exchange, and $A_s$ is a matrix quantifying the single-ion anisotropy. $H^{ani}$ breaks the SU(2) spin symmetry and fixes the global orientation of the ground state.

The magnetic Hamiltonian was fully parameterized from DFT calculations, as described in the section below. The values of both the isotropic and anisotropic interactions are found in Supplementary Table 1. Supplementary Note 2 discusses how our model compares with those previously reported in the literature[6,47].

### Density-functional theory calculations

We performed DFT calculations to obtain the exchange parameters and electric polarization, using the projector-augmented wave method as implemented in the VASP code. The Ni 3$d$ and 4$s$, as well as the I 5$s$ and 5$p$ orbitals, were explicitly included as valence states in the calculation, with a plane-wave cut-off of 350 eV. We used the Perdew–Burke–Erzenhof (PBE) exchange-correlation functional and performed PBE + $U$ calculation within the Dudarev approach, with $U = 4$ eV and the spin–orbit coupling fully considered.

To obtain the exchange parameters, we used the four-state method[48,49] on a $7 \times 5 \times 1$ supercell of monolayer $NiI_2$, and used a $k$-point mesh of $1 \times 1 \times 1$. We calculated the electric polarization using the modern theory of polarization for a $7 \times 1 \times 1$ supercell, compatible with the spin-spiral ground state, using a $k$-point mesh of $1 \times 6 \times 1$. To obtain the ground-state polarization, we calculated the polarization difference between the spin configurations with $\mathbf{q} = (1/7, 0, 0)$ and its mirror image with $\mathbf{q} = (-1/7, 0, 0)$. To estimate the electric dipole of the electromagnon modes, we displaced the equilibrium spin configuration according to the real-space structure of the magnon modes and calculated the change of polarization with respect to the equilibrium configuration. This formalism is also known as frozen-magnon approximation[21]. To constrain the spins into a given pattern, we used the penalty functional implemented in VASP with $\hbar\omega = 1.0$ eV.

### Ground-state and electromagnon polarization

The ground-state electric polarization was found to be $P_{el} = 5.3 \times 10^{-13}$ C m$^{-1}$ for the two-dimensional unit cell ($P_{el} \approx 8.0 \times 10^{-4}$ C m$^{-2}$ for the three-dimensional unit cell) and to be directed along the [010] direction, consistent with our symmetry analysis and previous works[15,17]. Similarly, the electromagnon mode $EM_o$ was found to have an electric dipole moment of magnitude $d_o = 2.5\mu_B/c$, perpendicular to the ground-state polarization, and the mode $EM_e$ to have an electric dipole moment of magnitude $d_e = 10.3\mu_B/c$ parallel to $P_{el}$. Here, $\mu_B$ is the Bohr magneton and $c$ is the speed of light. This shows that the modes are electromagnons with colossal oscillating electric dipole moments.

These results were further reproduced by an analytical generalization of the spin-current model presented in refs. 50,51 to the case of $NiI_2$. Using the generalized model, the change in electric polarization due to the magnetic order is given to the first order by

$$\mathbf{P} = \frac{\lambda t^3}{\Delta^4} d_{d-p} \mathbf{d}[\hat{\mathbf{n}} \cdot (\mathbf{S}_1 \times \mathbf{S}_2)], \tag{3}$$

where $\lambda$ is the I spin–orbit coupling, $t$ is the Ni–I hopping amplitude, $\Delta$ is the charge-transfer energy between Ni and I ions and $d_{d-p}$ is the dipole moment of an Ni–I bond. The vector $\hat{\mathbf{n}}$ is normal to the plane spanned by the Ni–I–Ni cluster containing spins $\mathbf{S}_1$ and $\mathbf{S}_2$, whereas the vector $\mathbf{d}$ is normal to both $\hat{\mathbf{n}}$ and the vector between the spins $\mathbf{S}_1$ and $\mathbf{S}_2$. Summing over the bonds of the magnetic unit cell yields a total electric polarization along the [010] axis of magnitude $P_{el} = 4.1 \times 10^{-13}$ C m$^{-1}$, in agreement with our first-principles calculations. For a detailed derivation, see Supplementary Note 2.

From equation (3), we find that for collinear spin structures or vanishing spin–orbit interaction, the electric polarization disappears. Therefore, a non-collinear magnetic order and spin–orbit interaction are necessary to induce a macroscopic electric polarization in $NiI_2$. Equation (3) also provides the electric polarization associated with each magnon mode, which agrees with our first-principles calculations and again confirms that these modes are electromagnons. We note that the strength of the electric polarization arises from the very large spin–orbit coupling of I, at a value of $\lambda \approx 0.5$ eV, as well as the unusual strength of the $d$–$p$ hybridization[22], defined as the ratio $t/\Delta \approx 0.33$.

### Data availability

All data required for assessing the conclusions are available online. Source data are provided with this paper.

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

**Acknowledgements** Work in the Baldini group at University of Texas at Austin was primarily supported by the Robert A. Welch Foundation under grant F-2092-20220331 (to F.Y.G. for data taking and analysis), the National Science Foundation under grant DMR-2308817 (to X.P. for data taking and analysis), and the Air Force Office of Scientific Research under Young Investigator Program award FA9550-24-1-0097 (to E.B. for data interpretation, manuscript writing and supervision). E.V.B. acknowledges funding from the Horizon Europe research and innovation program of the European Union under the Marie Skłodowska–Curie grant agreement no. 101106809. A.R. and E.V.B. acknowledge support from the Cluster of Excellence 'CUI: Advanced Imaging of Matter'—EXC 2056—project ID 390715994, and Grupos Consolidados (IT1453-22). A.R. and E.V.B. also acknowledge support from the Max Planck–New York City Center for Non-Equilibrium Quantum Phenomena. The Flatiron Institute is a division of the Simons Foundation. R.S. acknowledges financial support provided by the Ministry of Science and Technology in Taiwan under project nos. NSTC 111-2124-M-001-009, 110-2112-M-001-065-MY3 and Academia Sinica, project no. AS-iMATE-111-12. D.S.K. and X.L. acknowledge support from NSF DMR-1720595, DMR-2308817 for spectroscopy studies and the Air Force Office of Scientific Research under award number FA2386-21-1-4067 for bulk crystals, and the Welch Foundation Chair F-0014 for sample preparation. Part of the experiments were performed at the user facility supported by the National Science Foundation through the Center for Dynamics and Control of Material under cooperative agreement no. DMR-2308817 and the Major Research Instrumentation (MRI) programme DMR-2019130. Open access funding is provided by the University of Texas at Austin and the Max Planck Society.

**Author contributions** F.Y.G. and E.B. conceived the study and designed the research; F.Y.G. and X.P. performed the experiments and analysed the data under the supervision of E.B. and supported by D.S.K. and X.L.; F.Y.G., X.C. and E.V.B. performed theoretical analyses and interpreted the data under the supervision of M.A.S., T.K., P.T., A.R. and E.B.; R.K.J., D.V., K.R., R.S. and S.-F.L. grew the $NiI_2$ samples and performed X-ray diffraction and magnetic susceptibility characterization; F.Y.G., X.P., X.C., E.V.B. and E.B. prepared the paper with input from all other authors; E.B. supervised the project.

**Funding** Open access funding provided by Max Planck Society.

**Competing interests** The authors declare no competing interests.

**Additional information**
**Correspondence and requests for materials** should be addressed to Angel Rubio or Edoardo Baldini.

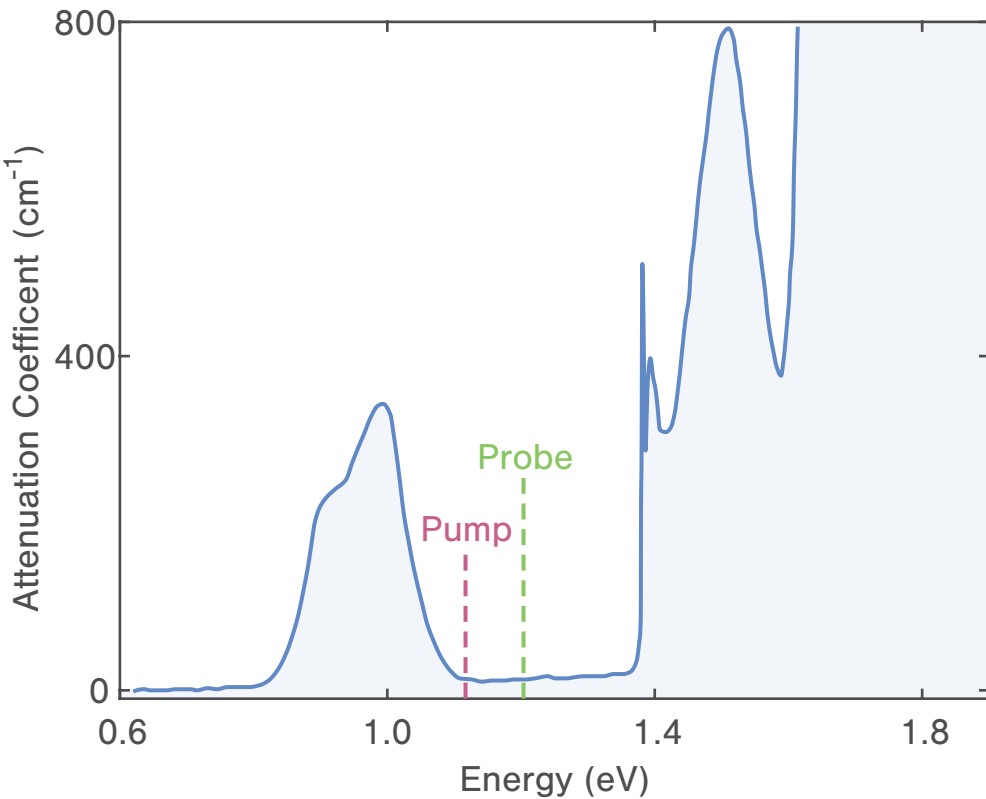

**Extended Data Fig. 1 | Optical absorption spectrum of NiI₂ at 5 K.** Light attenuation coefficient as a function of photon energy. A set of *d–d* transitions and exciton states emerge across the near-infrared spectral range. The vertical bars denote the photon energies used in our pump ($\hbar\omega_{pump}$ = 1.13 eV) and probe ($\hbar\omega_{probe}$ = 1.20 eV) experiment. Data is adapted from ref. 52.

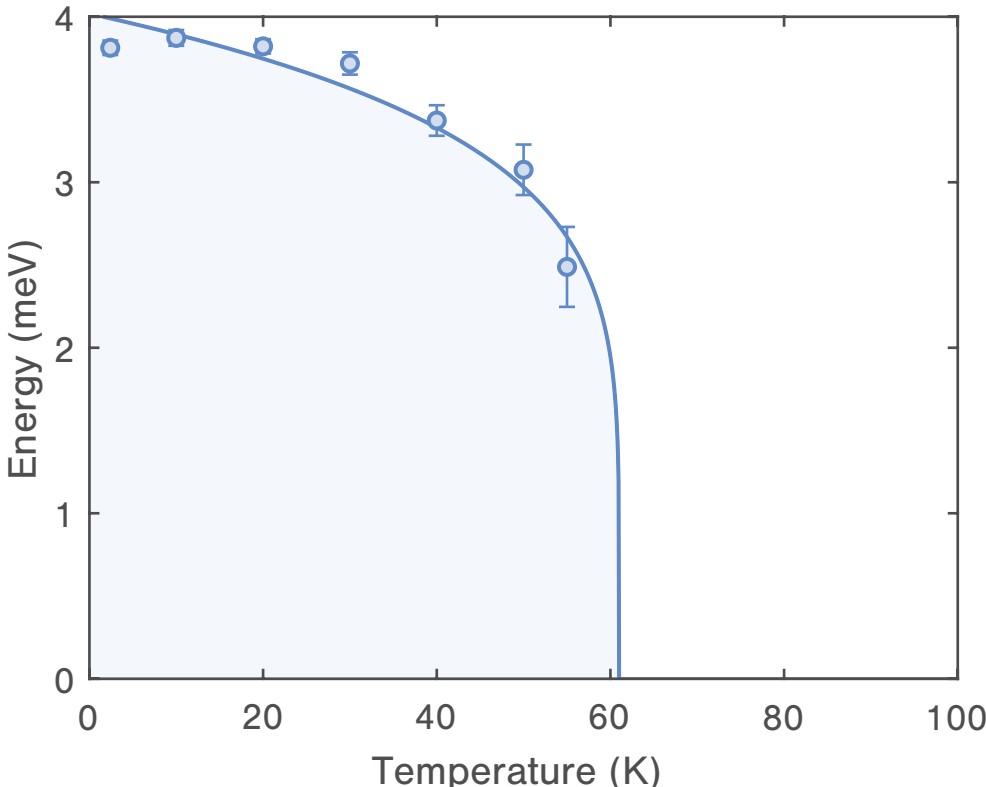

**Extended Data Fig. 2 | EM$_o$ energy in the tr-SHG data.** Temperature-dependent EM$_o$ energies obtained from fits of the tr-SHG data in Fig. 2b (*circles*) are shown alongside a guide to the eye (*line*).

**Extended Data Table 1 | Theoretical values of natural optical activity in resonance with electromagnon modes**

| Mode | $\alpha_{ij}$ | Value (ps m$^{-1}$) |
|------|---------------|---------------------|
| EM$_o$ | Im$[\alpha_{\kappa\kappa}]$ | $12 \times 10^3$ |
| EM$_e$ | Im$[\alpha_{yy}]$ | $49 \times 10^3$ |

Magnetoelectric coupling tensor elements $a_{ij}$ of the EM$_o$ and EM$_e$ electromagnon modes obtained from ab initio calculations within the frozen-magnon approximation. Note that the couplings of EM$_o$ and EM$_e$ are imaginary and diagonal along their respective modulations axes $\hat{\kappa} \sim (\hat{x} + \sqrt{2}\,\hat{z})/\sqrt{3}$ and $\hat{y}$, highlighting natural optical activity.