## [Peer Review File · Nature]

Manuscript Title: Giant chiral magnetoelectric oscillations in a van der Waals multiferroic

Editorial Notes:

Redactions – unpublished data

Reviewer Comments & Author Rebuttals

Reviewer Reports on the Initial Version:

Referees' comments:

Referee #1 (Remarks to the Author):

This manuscript reports the authors' investigation of NiI₂, a multiferroic insulator. The authors carry out theoretical calculations confirming the spiral magnetic order and the onset of ferroelectric polarization while discovering a record-high magnetoelectric coupling stemming from the frustrated magnetic order and the relativistic spin-orbit coupling. The reviewer finds this work quite interesting and well-organized to the extent that the condensed matter community can benefit from the new results reported here and the methodologies adopted here. If the following points are further clarified, the reviewer believes that the manuscript can be published in Nature.

1. The energies of the two electromagnons invoked here (3.85 meV and 4.27 meV) do not agree with what were reported in Ref. 17. The higher-energy mode reported here appears to match the lower-energy mode of Ref. 17. Do we expect differences here because Raman spectroscopy sees different electromagnon modes compared to terahertz (THz) spectroscopy? Perhaps, the discrepancy comes from incomplete modeling of the physics of NiI₂? All the theoretical results such as oscillation patterns and magnetoelectric coupling are affected by not properly accounting for the higher-energy mode seen in Ref. 17?

2. The NiI₂ flakes (please specify the thickness in the main text) may have been damaged during the laser excitations. Although the pump was done in non-resonant fashion, the small-sized domains might "dephase" upon encountering intense electric and magnetic fields of the pump beam. Have the authors re-check the status of the flake after the experiment? Parts of the single domain may have split into multi-domains during the heating of the sample.

3. Although one speaks of proper-screw helimagnetism in NiI₂, the spin-spiral planes are not exactly perpendicular to the q axis of the magnetic order. Is this fact taken into account during calculation and the ensuing estimate of the optical activity? The polarization also has components in-plane and out-of-plane. Are both of these well considered during calculation and experiment?

Referee #2 (Remarks to the Author):

In this work, the authors performed a precision measurement of the dynamical magnetoelectric coupling for an enantiopure domain in an exfoliated van der Waals multiferroic NiI₂. They revealed a record-breaking natural optical activity at terahertz frequencies, characterized by quadrature modulations between the electric polarization and magnetization components. They further performed first-principles calculations to show that these giant chiral couplings originate from the synergy between the non-collinear spin texture and relativistic spin-orbit interactions, resulting in substantial enhancements over lattice-mediated effects. They also developed a method to offer full tomography of the dynamical magnetoelectric tensor, disentangling different phenomena via the coherent response of collective modes. This study highlights the potential for intertwined orders to enable unique functionalities in the two-dimensional limit and pave the way for the development of van der Waals magnetoelectric devices operating at terahertz speeds. These results are interesting and I would like to recommend the publication of this work in Nature if the authors can address the following issues:

(1) In the paper by Song et al. [Nature 602, 601–605 (2022)], the spin order induced electric polarization in NiI₂ was calculated and analyzed with the general spin-current model [Phys. Rev. Lett. 107, 157202 (2011)]. The authors should compare the results with that obtained by Song et al. The current authors derived a generalization of the spin-current model, how does the generalization of the spin-current model differ from the general spin-current model proposed in Phys. Rev. Lett. 107, 157202 (2011)?

(2) The authors employed the four-state method to compute exchange interaction parameters. References to four-state method are needed.

(3) A realistic spin model for NiI₂ was constructed in Phys. Rev. Lett. 131, 036701 (2023). Is the model obtained by the current authors the same as or different from that model?

(4) Why does NiI₂ have a record-breaking natural optical activity at terahertz frequencies? What makes NiI₂ different from other multiferroics? Can the authors provide clue on how to enhance the chiral couplings?

Referee #3 (Remarks to the Author):

In the manuscript “Giant chiral magnetoelectric oscillations in a van der Waals multiferroic”, Gao et al present measurements of the dynamical magnetoelectric coupling in an exfoliated flake of the van der Waals multiferroic nickel iodide. The authors use a combination of pump-probe measurements and calculations to obtain the magnetoelectric coupling constant of this material, which is found to be on the order of 10 ns/m at 4 meV. This value is presented as the largest ever recorded.

This is an impressive piece of work, and is rather comprehensive in its nature. The authors have clearly expended a large amount of effort, and I must congratulate them on the results. They clearly show that the dynamical magnetoelectrical coupling in nickel iodide is remarkably strong, and the experimental scheme is very complete, allowing the material’s polarization and magnetization to be disentangled completely. The manuscript is very clearly written, and the results would be of topical interest to audiences including optoelectronics, ultrafast dynamics and spintronics.

There are a number of aspects that, I believe, should be addressed before allowing the manuscript to proceed to publication. I go into details below.

1. Novelty and breakthrough nature of the manuscript

1.1 The authors use first-principle calculations to extract, for nickel iodide, the magnetoelectric coupling strength in resonance to the two electromagnons. They obtain very large values on the order of 10 ns/m, which is attributed to the magnetic ordering and spin-orbit interaction. Then, the authors use pump-probe measurements to validate this theoretical prediction, obtaining a similar magnitude. On one hand, the excellent match between theory and experiment provides an open-and-shut case of a model being experimentally validated by experiment. On the other hand, publications in Nature should “represent an advance in understanding likely to influence thinking in the field”. The fact, therefore, that the very large value of the magnetoelectric coupling can already be described successfully by a contemporary model seems (to me) to leave little scope for future works based on this manuscript. Can the authors please describe how their measurements will influence thinking in the field, and what future challenges/opportunities are opened by their work? The line “Our study establishes strongly spin-orbit-coupled van der Waals materials as a marquee class of solids for achieving unparalleled dynamical magnetoelectric interactions” is too speculative, as nickel iodide seems to be the only van der Waals material known to date that has this large magnetoelectric coupling – I would argue that this work does not yet really reveal a “marquee class of solids”.

1.2 Connected to the above, Ref. 13 [Song et al, Nature 2022] has already identified that nickel iodide displays type II multiferroic order at the single layer limit. Moreover, Ref. 13 used circular dichroic Raman measurements and birefringence/SHG measurements in the static non-time-resolved scenario. Compared to Ref. 13, Gao et al use pump-probe measurements to extract the magnetoelectrical coupling constant of nickel iodide. The authors should mention this clarifying point in the manuscript, to make it much clearer that their work builds on the recent discovery published in Ref. 13 two years ago.

1.3 While the manuscript is very well written, I must admit that some of the lines seemed excessive. The examples are:

i) “To our knowledge, this is the first time the dynamical magnetoelectric coupling strength is measured in a microscale exfoliated flake”; Dynamical magnetoelectric couplings have been measured in a number of materials, as summarized by the authors themselves in Table S5. Does the point above really need to be emphasized?

ii) “the method pioneered in this work is poised to offer full tomography of the dynamical magnetoelectric tensor”; Pump-probe measurements are very routinely done to measure ultrafast dynamics of both magnetization or polarization in multiferroics. I therefore would argue that the method used by the authors, while elegant and impressive, is certainly not pioneered by the authors.

iii) “Our study establishes strongly spin-orbit-coupled van der Waals materials as a marquee class of solids for achieving unparalleled dynamical magnetoelectric interactions”; This is rather speculative, as nickel iodide seems to be the only van der Waals material known to date that has this large magnetoelectric coupling – I would argue that this work cannot yet reveal a “marquee class of solids”, since it studies only one material.

2. Experimental details

2.1 How smooth is the flake of nickel iodide under investigation? The authors used very advanced nanopositioners to control the sample position – are the signals highly dependent on where it is collected from the sample? Is the large magnetoelectrical coupling unique to a particular spot on the sample?

2.2 It is not completely clear how the sample was pumped. Given the presumably small working distance of the focusing objective lens, I suppose there is no space for the pump pulse to be directed to the sample at an angle. Was the pump therefore focused also by the objective lens? If so, how do the authors ensure that the probe pulse is smaller than the pump pulse on the sample surface (the standard requirement for pump-probe measurements)?

3. Scientific questions

3.1 The time-resolved measurements in Fig. 2b, d, f decay rapidly, on the timescale of 10 ps. Can the authors physically explain what governs this decay time? It is also not clear to me why the decay constant depends on the fluence (Fig. S10b). Perhaps I missed the explanation, but shouldn't the time constant be independent of the fluence?

3.2 The authors assess, in Supplementary Note 7, the pump polarization dependence. They find that the electromagnon oscillations are independent of the orientation of the linear polarization of the pump pulse. Do they find any different effect if instead using circularly-polarized pulses? This would be interesting from the point of view of identifying whether the inverse Faraday effect can be detected.

Author Rebuttals to Initial Comments:

Response to Reviews

Referee #1 (Remarks to the Author):

This manuscript reports the authors' investigation of NiI₂, a multiferroic insulator. The authors carry out theoretical calculations confirming the spiral magnetic order and the onset of ferroelectric polarization while discovering a record-high magnetoelectric coupling stemming from the frustrated magnetic order and the relativistic spin-orbit coupling. The reviewer finds this work quite interesting and well-organized to the extent that the condensed matter community can benefit from the new results reported here and the methodologies adopted here. If the following points are further clarified, the reviewer believes that the manuscript can be published in Nature.

We thank the Referee for appreciating our work and encouraging its publication in Nature.

1. The energies of the two electromagnons invoked here (3.85 meV and 4.27 meV) do not agree with what were reported in Ref. 17. The higher-energy mode reported here appears to match the lower-energy mode of Ref. 17. Do we expect differences here because Raman spectroscopy sees different electromagnon modes compared to terahertz (IR) spectroscopy? Perhaps, the discrepancy comes from incomplete modeling of the physics of NiI₂? All the theoretical results, such as oscillation patterns and magnetoelectric coupling, are affected by not properly accounting for the higher-energy mode seen in Ref. 17.

This comment by the Referee made us re-investigate the dataset acquired with spontaneous Raman scattering. We noticed that there was an error in the output of the acquisition software, which resulted in a slight offset in the reported Raman shifts and, thus, a slight shift in the observed collective mode frequencies. This error can be seen in Figure R1, where we have reproduced the uncorrected spontaneous Raman spectrum for the σ^+ channel (red) reported previously in Fig. 1d of the main text. The uncorrected spectrum shows several subtle discrepancies, including (1) the peak of the laser line not being aligned to 0 shift and (2) the very weak anti-Stokes peaks not having the same magnitude of Raman shift as their corresponding Stokes peaks. To correct these spectra, we recalculated the Raman shifts based on the known grating dispersion and aligned to the laser line peak, with the resulting corrected Raman signal (blue), also shown in Fig. R1. Doing so moves the electromagnon energies up by ~ 0.235 meV, yielding energies of 4.09 and 4.51 meV (33 and 36.4 cm^{-1}) for the two modes at 2.4 K. We thank the Reviewer for leading us to this discovery and have updated Fig. 1d with the corrected Raman spectra.

These energies are comparable to both the electromagnon modes observed previously in time-domain terahertz spectroscopy [J. H. Kim et al., Phys. Rev. B 108, 064414 (2023)], where the two electromagnon modes appeared at energies of 34 and 37 cm^{-1} at 1.5 K, and in spontaneous Raman scattering spectroscopy [W. Song et al., Nature 602, 601 (2022)], where the two modes were observed at 31 and 36 cm^{-1} at 30 K. Despite the slightly different frequencies at 30 K, the modes detected by time-domain terahertz spectroscopy and spontaneous Raman scattering must be the same because the material lacks inversion symmetry. Therefore, electric-dipole-

allowed zone-center collective modes are expected to exhibit both infrared and Raman activity. This interpretation is reinforced by the very similar temperature dependencies observed in the two cited papers.

The residual difference between the reported energies of our Raman measurements at 2.4 K (33 and 36.4 cm^{-1}) and the terahertz measurements at 1.5 K (34 and 37 cm^{-1}) is small and may be attributed to differences in the observed temperature or slight variations in the sample quality.

Fig. R1: Spontaneous Raman scattering spectra correction

2. The NiI_2 flakes (please specify the thickness in the main text) may have been damaged during the laser excitations. Although the pump was done in non-resonant fashion, the small-sized domains might “dephase” upon encountering intense electric and magnetic fields of the pump beam. Have the authors re-check the status of the flake after the experiment? Parts of the single domain may have split into multi-domains during the heating of the sample.

As requested by the Referee, we have added information about the thickness of our NiI_2 flakes to the main text (in addition to the methods previously). Regarding the impact of laser pumping, we recorded the SHG polarimetry signal after each pump-probe experiment and found no difference in the observed polar pattern. This, along with the fact that our time-resolved SHG signals were stable during each acquisition (~ 1 day), allows us to conclude that our pump-probe experiments did not cause any dephasing of the underlying domains and that the

measured enantiopure domain remained stable throughout the measurement. To address this comment, in our revised Supplementary Note 1, we have included a comparison between the static SHG polar pattern observed before laser excitation and the pattern acquired after 3 days of pump-probe experiments, see Fig. S3, as well as two scans of the time-resolved SHG polar patterns extracted from a pump-probe experiment that lasted an entire day, see Fig. S4.

3. Although one speaks of proper-screw helimagnetism in NiI_2 , the spin-spiral planes are not exactly perpendicular to the q axis of the magnetic order. Is this fact taken into account during calculation and the ensuing estimate of the optical activity? The polarization also has components in-plane and out-of-plane. Are both of these well considered during calculation and experiment?

We thank the Referee for this insightful question. Yes, in our calculations, we have considered the fact that the spin-spiral planes are not exactly perpendicular to the in-plane modulation vector \mathbf{q}_{in} . Experimentally, it is known that the bulk system displays a spin spiral structure with ordering vector $\mathbf{q} = (0.138, 0, 1.457)$ [S.R. Kuindersma et al., *Physica B+C* 111, 231-248 (1981)]. To account for this fact, we performed our density functional theory (DFT) calculations in the experimental magnetic structure. Subsequently, within our spin model, we fully relaxed the spin-spiral plane. With this approach, we found a canting angle of 45° between the spin-spiral and the monolayer planes, which differs from the angle found experimentally (55°). While all displayed results derived from the model were obtained using this relaxed configuration, we have checked that our calculated observables do not depend sensitively on the canting angle. This can be explained by the fact that all calculated quantities (*e.g.*, electromagnon energies, electric polarizations, magnetizations, and natural optical activities) are relative quantities, in the sense that they are obtained by considering small changes from a given equilibrium state. Such quantities are known to be less sensitive to parameter changes. Furthermore, the electric polarization can be shown to change with the canting angle as $P = P_0 \sin \theta$, where θ is the angle between \mathbf{q} and \mathbf{q}_{in} . This shows both that the change in polarization with respect to θ is approximately linear, such that small changes in θ lead to small changes in P , and that our theoretical results may slightly underestimate the polarization (since $\sin(45^\circ) < \sin(55^\circ)$). To address this point, we included an additional discussion in Supplementary Note 2 about the dependence of the magnetoelectric properties on the canting angle.

As for the electric polarization, we respectfully disagree with the Reviewer that this has both in-plane and out-of-plane components. In fact, due to the C_2 symmetry of the system, the electric polarization is enforced to lie along the C_2 axis [see *e.g.*, Q. Song et al., *Nature* 602, 601–605 (2022)]. In NiI_2 , this axis aligns within the monolayer plane and is perpendicular to the in-plane propagation vector \mathbf{q}_{in} . This symmetry argument is confirmed by both our DFT and spin model calculations. However, a small out-of-plane polarization does appear when the system is driven out-of-equilibrium by exciting the EM_o electromagnon mode, and this is fully accounted for by our calculations. To address the Reviewer's comment, we denoted the static electric polarization as "in-plane" throughout the main text.

Referee #2 (Remarks to the Author):

In this work, the authors performed a precision measurement of the dynamical magnetoelectric coupling for an enantiopure domain in an exfoliated van der Waals multiferroic NiI_2 . They revealed a record-breaking natural optical activity at terahertz frequencies, characterized by quadrature modulations between the electric polarization and magnetization components. They further performed first-principles calculations to show that these giant chiral couplings originate from the synergy between the non-collinear spin texture and relativistic spin-orbit interactions, resulting in substantial enhancements over lattice-mediated effects. They also developed a method to offer full tomography of the dynamical magnetoelectric tensor, disentangling different phenomena via the coherent response of collective modes. This study highlights the potential for intertwined orders to enable unique functionalities in the two-dimensional limit and pave the way for the development of van der Waals magnetoelectric devices operating at terahertz speeds. These results are interesting, and I would like to recommend the publication of this work in Nature if the authors can address the following issues:

We thank the Referee for appreciating our work and recommending its publication in Nature.

(1) In the paper by Song et al. [Nature 602, 601–605 (2022)], the spin order induced electric polarization in NiI_2 was calculated and analyzed with the general spin-current model [Phys. Rev. Lett. 107, 157202 (2011)]. The authors should compare the results with that obtained by Song et al. The current authors derived a generalization of the spin-current model, how does the generalization of the spin-current model differ from the general spin-current model proposed in Phys. Rev. Lett. 107, 157202 (2011)?

We thank the Referee for raising this point, as it allowed us to enhance the clarity of our manuscript. First, we note that our results are consistent with those obtained by Q. Song et al., Nature 602, 601–605 (2022). Importantly, Q. Song et al. found a large anisotropy of the polarization matrix M , which is captured by our analytical expression derived from a planar spin cluster. Correspondingly, the calculated value for the polarization, $P = 250 \times 10^{-5} \text{ e}\text{\AA}$ as reported by Song et al., compares well with our calculated polarization of $P = 230 \times 10^{-5} \text{ e}\text{\AA}$.

Second, we agree that the spin-current model proposed in H. J. Xiang et al., Phys. Rev. Lett. 107, 157202 (2011) is of the most general form. However, due to its complexity, one typically needs numerical methods to obtain the polarization matrix. Hence, it is valuable to have an analytic expression that provides insights into the physical properties and the parameters controlling the strength and direction of the electric polarization. For these reasons, our spin-current model is not a generalization of the work by H. J. Xiang et al., but instead a generalization of the analytical result obtained in H. Katsura et al., Phys. Rev. Lett. 95, 057205 (2005). In our revised manuscript, we have therefore clarified that our model is a generalization of the work by H. Katsura et al., not that of H. J. Xiang et al.

(2) The authors employed the four-state method to compute exchange interaction parameters. References to four-state method are needed.

As requested by the Reviewer, we have included the reference D. Šabani et al., Phys. Rev. B

102, 01 457 (2020), which discusses the four-state method in detail.

(3) A realistic spin model for NiI_2 was constructed in Phys. Rev. Lett. 131, 036701 (2023). Is the model obtained by the current authors the same as or different from that model?

The spin model considered in X. Li et al., Phys. Rev. Lett. 131, 036701 (2023) is identical to ours, up to differences in the values of the spin parameters. For comparison, Table R1 summarizes the spin parameters used in this work, the values reported in the work by X. Li et al., [REDACTED]

In our work, we employed the Perdew-Burke-Erzenhof (PBE) functional together with the four-state method to extract the spin parameters, while the work by X. Li et al. considers either the PBE or the hybrid HSE06 (HSE) functional together with a machine-learning approach. Interestingly, our values are closer to those obtained by X. Li et al. with the HSE functional rather than the PBE functional, indicating a dependence of the parameters both on the functional and on the method used to extract the parameters. However, since all parameter sets are in close agreement, both the electro magnon energies and the electric polarization are rather insensitive to the choice of functional and calculation method.

To support this statement, we note that the most sensitive property in our model calculations is the canting angle of the spin rotation plane with respect to the monolayer plane. In the work by X. Li et al., it was found that this angle depends sensitively on the ratio between the nearest-neighbor and third nearest-neighbor exchanges J_1 and J_3 , with a larger value of J_3/J_1 favoring a larger canting angle. As a result, in our model, we found an angle of 45° instead of the one of 55° measured experimentally. However, to compensate for this effect, we performed our DFT calculations in the experimental magnetic configuration and still obtained a similar value of the electric polarization as in our model (with $P_{\text{DFT}} = 5.3 \times 10^{-13}$ C/m and $P_{\text{model}} = 4.1 \times 10^{-13}$ C/m). Furthermore, from Eq. 1 below, the electric polarization can be shown to change with the canting angle as $P = P_0 \sin \theta$, where θ is the angle between the spiral propagation vector \mathbf{q} and its projection \mathbf{q}_{\parallel} onto the monolayer plane. This implies that the change in polarization with respect to θ is approximately linear (such that small changes in θ lead to small changes in P) and that our theoretical results likely underestimates the polarization (since $\sin(45^\circ) < \sin(55^\circ)$).

A similar argument can be made for the electromagnon energies. We note that the parameter set used in our work has slightly larger values of the nearest-neighbor and third nearest-neighbor exchanges J_1 and J_3 compared to the work by X. Li et al. In Fig. R2, we show the change in electromagnon energies when varying the third nearest-neighbor parameter J_3 and the Kitaev interaction K , keeping J_1 fixed to define the overall unit of energy (this parameter is in good agreement between different approaches, see Tab. R1). For moderate changes in the magnetic parameters, the electromagnon energies are approximately linear functions of the variation, showing that small changes in parameters lead to small changes in the energies (up to 10% for variation in J_3 or K up to 20%). In particular, our calculated electromagnon energies of 3.93 meV and 4.30 meV compare well with the experimental values of 4.09 meV and 4.51 meV, and would lie at even lower energies with the HSE parameters.

Based on these considerations, we are confident that our model gives a good description of the electromagnon properties and the associated ferroelectric properties, as validated by compar-

isons to both our DFT data and experiments. To address the Reviewer’s comments, we have added a thorough discussion in Supplementary Note 2, also including Fig. R2 and Table R1.

Fig. R2: Dependence of electromagnon energies on the third nearest-neighbor exchange parameter J_3 and the nearest-neighbor Kitaev interaction K . We find an approximately linear dependence of the electromagnon energies on both parameters, with changes in the electromagnon energies of up to 10% for changes in J_3 and K of up to 20%. The blue and red curves correspond to the C2-odd and C2-even electromagnons, respectively

[REDACTED]

(4) Why does NiI_2 have a record-breaking natural optical activity at terahertz frequencies? What makes NiI_2 different from other multiferroics? Can the authors provide clue on how to enhance the chiral couplings?

The giant natural optical activity at terahertz frequencies can be attributed to the large electric polarization exhibited by the electromagnons in NiI_2 . According to our first-principles calculations and the effective spin model, the electromagnon polarization at terahertz frequencies can

be expressed as

$$\hat{\mathbf{P}}^{(1)} = \frac{\lambda t^3}{\Delta^4} d_{d-p} (\hat{\mathbf{e}}_1 + \hat{\mathbf{e}}_2) [\hat{\mathbf{e}}_3 \cdot (\mathbf{S}_1 \times \mathbf{S}_2)]. \quad (1)$$

Here λ is the spin-orbit coupling on the I atoms, t is the Ni-I hopping amplitude, Δ is the charge-transfer energy between Ni and I atoms, and d_{d-p} is the dipole moment of a Ni-I bond. The large natural optical activity therefore has three main microscopic origins.

First, we find that the electric polarization and nonlinear optical susceptibility are both proportional to the spin-orbit coupling on the I ligands. As I is a heavy element, it has a very large spin-orbit coupling ($\lambda \approx 0.5$ eV).

Second, since the electromagnon polarization in NiI_2 arises from the inverse Dzyaloshinskii–Moriya interaction (DMI), and not from phonon-mediated mechanisms as in many other multiferroics, the effective dynamical magnetoelectric coupling is enhanced compared to such materials by a factor $1/g^2$ (with g the electron-phonon coupling).

Finally, we find that the electric polarization and nonlinear optical susceptibility depend sensitively on the $d-p$ hybridization between the Ni ions and I ligands. Since Ni has a large electronegativity among $3d$ transition metals and I has a large atomic radius, the $d-p$ hybridization in NiI_2 is unusually strong [see also K. Riedl et al., Phys. Rev. B 106, 035156 (2022)]. To make this statement quantitative, we explicitly calculated the Wannier orbitals of NiI_2 , from which we found the $d-p$ hybridization strength (defined as the ratio t/Δ) to be around $1/3$. Since the factor t/Δ enters to the third power in Eq. 1, the strong $d-p$ hybridization results in a greatly increased dynamical magnetoelectric coupling compared to most materials.

In conclusion, our analysis suggests that the large dynamical magnetoelectric coupling in NiI_2 arises from a synergistic interplay of several beneficial properties, namely the strong spin-orbit coupling on I, the inverse DMI mechanism, and the strong $d-p$ hybridization. To enhance the chiral coupling, we therefore propose to search for materials in which the spin-orbit coupling and $d-p$ hybridization can be enhanced. Since many van der Waals materials show similar atomic configurations as NiI_2 , with competing magnetic interactions that could stabilize a non-collinear magnetic order, we believe that these systems are promising candidates. To address the Reviewer’s comment, we have modified the Discussion section in our main text and expanded Supplementary Note 2.

Referee #3 (Remarks to the Author):

In the manuscript “Giant chiral magnetoelectric oscillations in a van der Waals multiferroic”, Gao et al present measurements of the dynamical magnetoelectric coupling in an exfoliated flake of the van der Waals multiferroic nickel iodide. The authors use a combination of pump-probe measurements and calculations to obtain the magnetoelectric coupling constant of this material, which is found to be on the order of 10 ns/m at 4 meV. This value is presented as the largest ever recorded.

This is an impressive piece of work, and is rather comprehensive in its nature. The authors have clearly expended a large amount of effort, and I must congratulate them on the results. They clearly show that the dynamical magnetoelectrical coupling in nickel iodide is remarkably strong, and the experimental scheme is very complete, allowing the material’s polarization and magnetization to be disentangled completely. The manuscript is very clearly written, and the results would be of topical interest to audiences including optoelectronics, ultrafast dynamics and spintronics.

There are a number of aspects that, I believe, should be addressed before allowing the manuscript to proceed to publication. I go into details below.

We thank the Reviewer for their kind words and their support of our manuscript for publication in Nature.

1. Novelty and breakthrough nature of the manuscript

1.1 The authors use first-principle calculations to extract, for nickel iodide, the magnetoelectric coupling strength in resonance to the two electromagnons. They obtain very large values on the order of 10 ns/m, which is attributed to the magnetic ordering and spin-orbit interaction. Then, the authors use pump-probe measurements to validate this theoretical prediction, obtaining a similar magnitude. On one hand, the excellent match between theory and experiment provides an open-and-shut case of a model being experimentally validated by experiment. On the other hand, publications in Nature should “represent an advance in understanding likely to influence thinking in the field”. The fact, therefore, that the very large value of the magnetoelectric coupling can already be described successfully by a contemporary model seems (to me) to leave little scope for future works based on this manuscript. Can the authors please describe how their measurements will influence thinking in the field and what future challenges/opportunities are opened by their work? The line “Our study establishes strongly spin-orbit-coupled van der Waals materials as a marquee class of solids for achieving unparalleled dynamical magnetoelectric interactions” is too speculative, as nickel iodide seems to be the only van der Waals material known to date that has this large magnetoelectric coupling – I would argue that this work does not yet really reveal a “marquee class of solids”.

We thank the Reviewer for their insightful comment. While it may seem that our results leave little scope for future works because of the excellent agreement between experiment and theory, we want to remark that both our experimental methodology and theoretical calculations represent state-of-the-art approaches, hitherto unexplored to our knowledge. Experimentally,

the measurement of the electric polarization and magnetization dynamics of coherent electromagnons within individual multiferroic domains, particularly in a two-dimensional material flake, is unprecedented. Theoretically, our work marks the first instance where an *ab initio* theory describes the nonequilibrium dynamics of magnon modes and their magnetoelectric coupling in a spiral magnet. Though we did not extensively emphasize these aspects in our paper to keep the language appropriate to the journal, we firmly believe that our approach heralds new avenues for the exploration of multiferroics, and we are confident that the community will appreciate these aspects even without explicit remarks. In response to the Referee's feedback, we have decided to adopt a different approach by restructuring the Discussion and Conclusions sections. In the Discussion section, we now highlight how our findings identify the essential components for achieving giant dynamical magnetoelectric coupling strength in type-II multiferroics. This section will be useful for those scientists who are interested in the design and discovery of new materials with large terahertz natural optical activity. In the Conclusion section, we briefly delineate the opportunities and challenges that are opened by our research. In particular, we focus on novel experimental avenues such as chiral domain imaging via the protocol that we demonstrated and on-chip chiral terahertz platforms integrating van der Waals multiferroics. Moreover, we describe future theoretical challenges, emphasizing the need to incorporate mutual nonlinear couplings between electromagnon modes to demonstrate chiral domain switching induced by tailored terahertz fields. Finally, we mention the importance of including strong light-matter coupling terms to explore the fate of the giant chiral magnetoelectric coupling in the context of chiral optical cavities, specifically in the creation of electromagnon-polariton states and in their impact on the magnetic ground state. The length of these sections were made compatible with the length restrictions typical of Nature articles. Still, we believe that this comment by the Reviewer has allowed us to strengthen the final portion of our article.

1.2 Connected to the above, Ref. 13 [Song et al, Nature 2022] has already identified that nickel iodide displays type II multiferroic order at the single layer limit. Moreover, Ref. 13 used circular dichroic Raman measurements and birefringence/SHG measurements in the static non-time-resolved scenario. Compared to Ref. 13, Gao et al use pump-probe measurements to extract the magnetoelectrical coupling constant of nickel iodide. The authors should mention this clarifying point in the manuscript, to make it much clearer that their work builds on the recent discovery published in Ref. 13 two years ago.

As requested by the Reviewer, we have modified the introduction to properly account for the discovery of the electromagnons. As in our paper we rely on both the Raman and hyper-Raman properties of these modes, we first included all the relevant works that measured the electromagnons of NiI₂ in equilibrium conditions (with spontaneous Raman scattering or terahertz spectroscopy). We then put emphasis on the study by Q. Song et al., highlighting that this paper discovered the electromagnons' finite Raman circular dichroism that provided the starting point for our work.

1.3 While the manuscript is very well written, I must admit that some of the lines seemed excessive. The examples are:

i) "To our knowledge, this is the first time the dynamical magnetoelectric coupling strength is measured in a microscale exfoliated flake"; Dynamical magnetoelectric couplings have been

measured in a number of materials, as summarized by the authors themselves in Table S5. Does the point above really need to be emphasized?

We agree with the Reviewer. In our revised version, we removed the sentence highlighted above.

ii) “the method pioneered in this work is poised to offer full tomography of the dynamical magnetoelectric tensor”; Pump-probe measurements are very routinely done to measure ultra-fast dynamics of both magnetization or polarization in multiferroics. I therefore would argue that the method used by the authors, while elegant and impressive, is certainly not pioneered by the authors.

We have also removed that statement to avoid the idea that this method was pioneered in this work.

2. Experimental details

2.1 How smooth is the flake of nickel iodide under investigation? The authors used very advanced nanopositioners to control the sample position – are the signals highly dependent on where it is collected from the sample? Is the large magnetoelectrical coupling unique to a particular spot on the sample?

To address the Reviewer’s first comment, we modified Supplementary Note 1 and included the atomic force microscopy (AFM) characterization of a typical NiI₂ flake. The measurements, shown in Fig. S2, allowed us to estimate that the sample is very smooth, exhibiting variations of only 0.30 nm at the exposed surface.

Regarding the Reviewer’s questions on the collection position, we find that the signals are qualitatively consistent across different locations on the sample, but they present differences that can be highlighted only by the use of our microscopy mode (with a spot size of 1 μm).

First, even in high-quality (lab-grown) NiI₂ samples, the electromagnon modes show a distribution of energies that highly depend on the sample location. This can be observed by measuring the spontaneous Raman microscopy response in different parts of the same flake (see Table R2). Such a distribution originates from the sensitivity of the electromagnons’ energies to the local single-ion anisotropy, which depends sensitively on the local environment (including the presence of domain walls and defects). **[REDACTED]** highlighting the need for microscopy probes – like in the present study – to extract accurate and intrinsic information from single domains.

Sample spot	EM _o energy (meV)	EM _e energy (meV)
1	3.68	4.44
2	3.68	4.10
3	3.33	4.30
4	3.61	4.23
5	3.40	4.37
6	3.82	4.27
7	4.10	4.51
8	3.68	4.51
9	3.54	4.30

Table R2: Distribution of electromagnon energies

Secondly, without a proper alignment with our nanopositioners, the domain structure in native NiI₂ would be highly complex. Scanning a typical sample reveals SHG polar patterns corresponding to a mixture of domain orientations with different electric polarization directions. These sample locations are not suitable for conducting accurate pump-probe experiments because the electromagnon oscillations originating from different domains can cancel each other out (through π phase shifts caused by distinct chiralities) and lead to a reduced value of magnetoelectric coupling.

Despite these differences, when we focus our beams within a high-quality enantiopure domain, we observe comparable signatures of electromagnon oscillations in both the tr-SHG and tr-RKerr channels, with the extracted magnetoelectric coupling constant being largely consistent. An example can be seen in Fig. R3, which shows a measurement on a different enantiopure domain in NiI₂. The data are consistent with the ones reported in the main text. For the published measurements, we chose spots on the sample that show the highest electromagnon energies (according to Raman) and, thus, are the least impacted by defects or domain boundaries.

Fig. R3: Tr-SHG on an additional enantiopure domain

2.2 It is not completely clear how the sample was pumped. Given the presumably small working distance of the focusing objective lens, I suppose there is no space for the pump pulse to be directed to the sample at an angle. Was the pump, therefore, focused also by the objective lens? If so, how do the authors ensure that the probe pulse is smaller than the pump pulse on the sample surface (the standard requirement for pump-probe measurements)?

The Reviewer is correct in that there is no space for the pump beam to be directed to the sample at an angle. In our time-resolved experiments, the pump and probe beams were combined with a beamsplitter. They propagated collinearly into the objective and overlapped on the sample. To ensure that the pump spot size was larger than the probe spot size, we relied on the fact that our probe photon energy (1.20 eV) fell within the achromatic range of our objective (1.16-2.67 eV), whereas our pump photon energy (1.13 eV) did not. Thus, the probe was tightly focused into a diffraction-limited spot of 1 μm onto the sample, while the pump spot size remained poorly focused and relatively large ($\sim 2.5 \mu\text{m}$). This approach allowed us to create a uniform photoexcited region to be probed. To address this comment, we added text to our Methods section.

3. Scientific questions

3.1 The time-resolved measurements in Fig. 2b, d, f decay rapidly, on the timescale of 10 ps. Can the authors physically explain what governs this decay time? It is also not clear to me why

the decay constant depends on the fluence (Fig. S10b). Perhaps I missed the explanation, but shouldn't the time constant be independent of the fluence?

We agree that the linear increase in the decay rate merits consideration. In the following, we will provide an overview of these decay processes.

First, we note that observations of a linear increase in the damping rate of a coherent collective mode have been frequently reported in the literature for both coherent phonons [K. Yee et al., Phys. Rev. Lett. 88, 105501 (2002); A.Q. Wu et al., Appl. Surf. Sci. 253, 6301 (2007); A.Q. Wu et al., Appl. Phys. Lett. 92 (2008); T.-Y. Jeong et al., J. Kor. Phys. Soc. 73, 951 (2018)] and magnons [De et al. Phys. Rev. B 109, 024422 (2024)]. The decay rate of these coherent modes, Γ , can be expressed as the sum of a population relaxation rate Γ_1 and a pure dephasing rate Γ_2 :

$$\Gamma = \frac{\Gamma_1}{2} + \Gamma_2. \quad (2)$$

where both Γ_1 and Γ_2 can be fluence-dependent parameters.

For pure dephasing processes, theoretical analyses [H. Yuan et al., Phys. Rev. B 106, L100403 (2022)] and experimental data [D. Bossini et al., Nat. Commun. 7, 10645 (2016)] have shown that Γ_2 can govern the decay dynamics of coherent magnons through mutual scattering events on the same magnon branch. Such magnon scattering would scale with the density of photoexcited coherent magnons, leading to an increase in the overall damping rate with increasing pump fluence.

The other possible damping terms are population relaxation processes. In particular, the dominant channels for coherent optical phonons and non-collinear antiferromagnetic magnons are anharmonic decay and three/four-magnon scattering processes, respectively. These decay processes scale with the population of incoherent phonons/magnons in the material, which can be induced by the optical pump through residual absorption. For instance, in a three-particle scattering process where a zone-center optical phonon/magnon with frequency ω_0 scatters into a pair of acoustic modes with opposite crystal momentum \mathbf{q} and $-\mathbf{q}$ and frequency $\omega_0/2$, the population relaxation rate Γ_1 can be described as:

$$\Gamma_1 = \Gamma_1^0(1 + n_{\mathbf{q}} + n_{-\mathbf{q}}), \quad (3)$$

where Γ_1^0 is the intrinsic decay rate at zero temperature, and $n_{\mathbf{k}}$ represents the phonon/magnon population of the scattering states. Assuming that the photoexcited incoherent phonon/magnon population scales linearly with the pump fluence (*i.e.*, the linear pump-probe regime), this will explain the observed linear increase in the damping rate [De et al. Phys. Rev. B 109, 024422 (2024)]. Alternatively, electromagnons could also scatter with charge carriers generated by the residual absorption of the optical pump pulse [K. Yee et al., Phys. Rev. Lett. 88, 105501 (2002)]. This process is frequently invoked in the existing literature on coherent optical phonons to explain similar observations of linear increases in the damping rate with pump fluence [A.Q. Wu et al., Appl. Surf. Sci. 253, 6301 (2007); A.Q. Wu et al., Appl. Phys. Lett. 92 (2008); T.-Y. Jeong et al., J. Kor. Phys. Soc. 73, 951 (2018)].

Disentangling the precise contributions of all the above factors to the overall decay dynamics

of an electromagnon will be a fruitful but challenging task that requires considerable additional theoretical and experimental efforts. Such an analysis, therefore, falls outside the scope of the current work and should merit its standalone publication. For this article, we summarized these considerations in our modified Supplementary Note 5.

3.2 The authors assess, in Supplementary Note 7, the pump polarization dependence. They find that the electromagnon oscillations are independent of the orientation of the linear polarization of the pump pulse. Do they find any different effect if instead using circularly-polarized pulses? This would be interesting from the point of view of identifying whether the inverse Faraday effect can be detected.

We thank the Reviewer for the insightful question. Based upon this suggestion, we decided to perform a thorough investigation of the induced electromagnon oscillations as a function of the incident pump polarization (including circularly polarized). The new data is shown in Fig. S16. There, we show that the electromagnon amplitude and phase do not depend on the incident linear pump polarization. Minor changes on the order of $< 10\%$ are likely due to the natural birefringence of the material. Similarly, the electromagnon amplitude and phase are invariant when the pump helicity is switched between left and right. Altogether, these results strongly indicate that the generation of the coherent electromagnon modes cannot be explained via the inverse Cotton-Motton or inverse Faraday effects. Combining these observations, we believe that the likely excitation mechanism is mediated by the photoexcited carrier density produced by the residual absorption of the 1.13 eV optical pump pulse. Recent studies have identified possible microscopic mechanisms that can lead to such results in various canted and zig-zag antiferromagnets [R.V. Mikhaylovskiy et al., Nat. Commun. 6, 8190 (2015); C.A. Belvin et al., Nat. Commun. 12, 4837 (2021); S. Toyoda et al., Phys. Rev. B 109, 064408 (2024); C.J. Allington et al., arXiv:2402.17041 (2024).] Some of these findings have been obtained by driving materials in the vicinity of $d-d$ transitions, similar to our case. Possible extensions of these theories to the case of spiral magnetic orders that lead to type-II multiferroicity may shed light on the microscopic details of the mechanism at play in NiI₂. To address the Reviewer's comment, we rewrote Supplementary Note 7.

Reviewer Reports on the First Revision:

Referees' comments:

Referee #1 (Remarks to the Author):

The authors have positively and conclusively responded to my previous comments and criticism. With some corrections to their previous data, the experimental observations are now quite consistent with previous two reports. This was the most serious concern in the first round of review. Secondly, they clarified that the ultrafast probes have not seriously damaged the samples during measurement, which substantiates their picture of the underlying physics and theoretical analysis. The reviewer believes that the manuscript is now suitable for publication in Nature.

Referee #2 (Remarks to the Author):

I would recommend the publication of this work after the following issues are addressed:

(1) In my opinion, the current authors derived a generalization of the spin-current model which should give the same results as

the general spin-current model proposed in Phys. Rev. Lett. 107, 157202 (2011). This point should be mentioned and this work should be cited.

(2) The four-state method was originally proposed in Phys. Rev. B 84, 224429 (2011) and reviewed in Dalton Trans., 2013,42, 823-853. These references should be cited.

Referee #3 (Remarks to the Author):

The authors have presented a thorough and convincing reply to all the comments of the three reviewers. I have read through the response, revised manuscript and supplementary information, and find that the manuscript has certainly improved and benefitted from the reviews. Overall, I am now happy to recommend publication of the manuscript in Nature.

Author Rebuttals to First Revision:

Response to Reviews

Referee #1 (Remarks to the Author):

The authors have positively and conclusively responded to my previous comments and criticism. With some corrections to their previous data, the experimental observations are now quite consistent with previous two reports. This was the most serious concern in the first round of review. Secondly, they clarified that the ultrafast probes have not seriously damaged the samples during measurement, which substantiates their picture of the underlying physics and theoretical analysis. The reviewer believes that the manuscript is now suitable for publication in Nature.

We thank the Referee for their insightful comments regarding our manuscript and for supporting its publication in Nature.

Referee #2 (Remarks to the Author):

I would recommend the publication of this work after the following issues are addressed:

(1) In my opinion, the current authors derived a generalization of the spin-current model which should give the same results as the general spin-current model proposed in Phys. Rev. Lett. 107, 157202 (2011). This point should be mentioned and this work should be cited.

(2) The four-state method was originally proposed in Phys. Rev. B 84, 224429 (2011) and reviewed in Dalton Trans., 2013,42, 823-853. These references should be cited.

Many thanks to the Referee for supporting the publication of our work in Nature. We now cite all three pertinent articles in the main text and supplementary information of our manuscript.

Referee #3 (Remarks to the Author):

The authors have presented a thorough and convincing reply to all the comments of the three reviewers. I have read through the response, revised manuscript and supplementary information, and find that the manuscript has certainly improved and benefitted from the reviews. Overall, I am now happy to recommend publication of the manuscript in Nature.

We appreciate the Referee's thoughtful feedback on our manuscript and their recommendation for publication in Nature.